# OpenReview forum: "Open Sesame! Universal Black Box Jailbreaking of Large Language Models"
_ICLR.cc/2024/Conference — Submitted to ICLR 2024_

### Official Review · Reviewer_1qTf · 2023-10-22

**Soundness:** 3 good
**Presentation:** 3 good
**Contribution:** 2 fair
**Rating:** 5
**Confidence:** 4

**Summary:**

While LLMs are designed to provide safe and helpful responses, they can be manipulated for malicious purposes due to their alignment techniques. This paper introduces a genetic algorithm-based approach to manipulate LLMs even when their internal architecture and parameters are hidden. The motivation lies in the desire to expose these vulnerabilities and understand the limitations of LLMs by providing a diagnostic tool to assess and enhance their alignment with human intent.

**Strengths:**

The paper introduces an approach to manipulate LLMs in a black-box scenario. Many users and organizations interact with LLMs in a black-box manner. This approach has the potential to reveal vulnerabilities in LLMs.

The paper provides a thorough and well-structured account of how GA are utilized for the black-box jailbreaking of LLMs. This meticulous methodology description allows readers to gain a clear understanding of the research process, from the use of GAs for optimization to the formulation of fitness functions and the evaluation of attack success rates.

The use of publicly available datasets enhances the transparency and reproducibility of the research.

**Weaknesses:**

While the paper outlines the use of GA to optimize the universal adversarial prompt, it seems does not provide sufficient information on the initial selection of the adversarial suffix that constitute the core component of the method.

Given the potential ethical implications of the research, consider including more details about the ethical considerations and safeguards implemented to prevent misuse of the research findings.

Minor Issues:
The reference should have been revised. Some preprints have been published, e.g., '(Certified!!) Adversarial Robustness for Free!' is published on ICLR 2023.

**Questions:**

Is there any existing work that can be compared? Like comparing to Zou et al.(2023) in the white-box case.

LLaMA2-7b has a higher SR than Vicuna-7b. I'm curious what causes this difference, training/fine-tuning method? Dataset? It might be helpful to understand the Transferability.

---

> ### Author Response · Authors · 2023-11-21
> **Responses and fixes to our paper following referee reports**
>
> Dear Reviewer,
>
> We sincerely appreciate your constructive feedback and valuable suggestions.
>
> Following the reports by yourself and the other reviewers we performed numerous runs over the past few weeks and have many additional results, which we've added to the paper.
>
> We have revised our manuscript according to your comments.
>
> We acknowledge the areas you've highlighted that require our attention and have attempted to address each of your comments as follows:
>
> Regarding your comments:
> * We have used initialization at random because it proved sufficient.
> * We have added an appendix in the revised paper regarding ethical considerations and safeguard implements to prevent misuse of the research findings.
> * We have fixed all the references in the paper according to your comment.
>
> Questions:
> * There are no known black-box attacks. As for white box: We would rather not compare white-box to black-box attacks since they have *very* different threat models, and this seems to us like an apples-and-oranges comparison.
> * We don’t know the actual reason for this but we think that the RLHF used in LLaMA2-7b is better in terms of “fulfilling the user’s request” than the supervised instruction tuning used in Vicuna. We have used the same dataset and same train/set split. We have tried different configurations to see the differences between these two models.
> * We have tried transferability to ChatGPT but it failed. It would seem that OpenAI has strengthened ChatGPT in response to the Zou et al. paper. This cat-and-mouse game is to be expected. That said, we added many more experiments with 13B models, which we were able to run with our given academic resources.
> * We have clarified all the above in the revised version.
>
> We hope these revisions address your concerns and strengthen the paper.
>
> We've uploaded a revised version.
>
> Thanks!

---

> > ### Comment · Reviewer_1qTf · 2023-11-22
> >
> > Thanks to the authors for clarifying my confusion. I am keeping my score unchanged for now.

---

> > > ### Author Response · Authors · 2023-11-22
> > >
> > > Thank you. We emphasize again that we've gotten a lot of new results and fixed all comments 😊

---

### Official Review · Reviewer_HedS · 2023-10-27

**Soundness:** 3 good
**Presentation:** 2 fair
**Contribution:** 2 fair
**Rating:** 5
**Confidence:** 4

**Summary:**

This paper proposes the use of genetic algorithms to jailbreak black box large language models (LLMs) in an automated way.  The authors provide an optimization-based formulation, and describe the steps of their algorithm in detail.  They also conduct a set of experiments using the harmful behaviors dataset to demonstrate the effectiveness of their method.


**Summary of my rating.**  Based on the discussion in this review, my initial assessment is as follows.  This paper is among the first to propose an automated black box jailbreak for LLMs.  The setting is of interest, the authors can claim novelty in their problem selection.  The empirical results also seem quite strong.  However, the problem formulation and notation are somewhat loose/non-rigorous, the paper lacks comparisons to baseline algorithms (which they open the door to earlier on in the paper), the experiments admit methodological concerns, particularly WRT the training and test split, the discussion of related work is sparse, and the conclusions don't necessarily align with the evidence.  For this reason, despite the novelty of this approach, my assessment is that this paper is not yet ready for publication, and could be significantly strengthened in a subsequent draft.

**Strengths:**

**Black box setting.**  This paper focuses on black-box attacks.  This setting is of particular interest, especially nowadays, since many popular LLMs are only available as black boxes.  If methods can be designed that reliably jailbreak black box LLMs, and particularly if they are relatively query efficient, this paper could have strong impact.

Jailbreaking LLMs is also a relatively new problem setting, and the authors are correct that they are among the first to propose a black box algorithm for accomplishing this task.  This constitutes novelty, and will factor into my overall assessment of this paper.

**Experimental results.** At face value, the empirical results are quite strong.  The authors report ASRs in the mid-ninety-percent range for both Llama2 and Vicuna.  These numbers are notably stronger than white-box attacks like GCG (although the experimental set up is different, as discussed in more detail below).  The authors also report relatively strong transferability in the appendix.

**Writing.** The paper is generally free of typos and grammatical errors.

**Weaknesses:**

### Problem formulation
---

The problem formulation could benefit from a rewrite.  The notation is relatively loose/non-standard, and the writing could be polished.  Here are some comments:

* In the equation following “Hence, the universal attack. . .” on page 5, the optimization variable is not explicitly included in the objective, which will likely confuse readers.
* The authors use the words “lexicon” and “vocabulary” to refer to the same set.
* The fitness function $\mathcal{L}$ is defined in two different ways
* Calligraphic notation is used for functions and for integers.  Consider using lower-case letters, e.g., $n$ and $m$, for your hyper parameters and calligraphic fonts for your losses, e.g., $\mathcal{L}
* It would be clearer if the authors used equation numbers.
* The algorithms are written out in words.  It may be clearer if the authors use the notation used in the paper to write out the steps of the algorithm.

### Lack of experimental evidence
---

**Comparisons to GCG.** Regarding comparisons to GCG, it’s arguable that as the proposed algorithm is meant for the black-box settings whereas GCG is a white-box attack, a comparison is not needed.  However, there are a few reasons why I think a comparison to GCG is necessary for this work.

Firstly, while discussing their past work, the authors say the following:

> “Our black box approach does not rely on a model’s internals, and thus we do not need to deal with these kinds of difficulties. Other recent works of ours have focused on leveraging evolutionary algorithms in black box scenarios, showing that we can match or surpass white box results, without recourse to model internals, and using no costly gradient descent.”

If a comparison between white-box and black-box algorithms had not been mentioned, one could make a stronger argument regarding why the comparison is not needed.  However, since the authors opened the door to this, and note that they offered such a comparison in past work, it seems reasonable to expect such a comparison here.

Secondly, although GCG is a white-box attack, the authors of the GCG paper show that attacks optimized for Vicuna and Llama2 can be transferred to black box LLMs, resulting in non negligible ASRs.  In this way, one could view GCG as a black box attack for models like GPT, Claude, and PaLM when the suffixes are transferred from Llama2 or Vicuna.

**Black-box attacks.**  One of the main arguments in the paper is that the proposed attack works in the black-box setting, wherein the attacker doesn’t have access to the parameters of the LLM.  The authors motivate this in the following way:

> “Though successful, the attack is limited because due to its white-box nature, meaning full access to the targeted model, including architecture, gradients and more. Such access is often not granted in real life.”

Given this, one would certainly expect to see experiments on LLMs that are exclusively available as black boxes, i.e., models that are only available via API calls (e.g., ChatGPT, Bard, Claude, etc.).  However, the experiments are only performed on open-source LLMs—specifically, Llama2 and Vicuna.  The paper feels incomplete without these experiments.

**Concerns about performance.**  The scores reported by the authors are impressive — they claim to be able to achieve ASRs in the high 90% range for both Llama2 and Vicuna.  However, I think that more context is needed before these results are publishable.  Specifically, how was the ASR measured here?  That is, how did the authors determine whether or not a jailbreak occurred?  In the GCG paper, the authors check for various strings (e.g., “I’m sorry” or “As an AI language model”) in the response.  Was the same criteria used here?  If the authors used the same method as in the GCG paper, then the results would indicate that this black-box attack is *significantly* more effective than the white-box GCG attack, which is unintuitive, given that GCG has more information about the network at its disposal.

**Repeats.**  The authors split the harmful behaviors dataset into training and test subsets.  However, it’s arguable as to whether this split makes sense, especially because harmful behaviors contains numerous duplicates.  For instance, there are several dozen queries that—using nearly identical language—ask the LLM to generate instructions detailing how one would build a bomb.  Therefore, there is undoubtedly some mixing between the training and test set, and so the extent to which we should consider these attacks to be universal is questionable.

### Paper feels unfinished
---

**Is this paper finished?**  This paper feels unfinished.  Aside from stating their results for direct and transferred attacks, there are no other experiments.  One would expect to see detailed ablation studies, including (1) testing different embedding functions, (2) measuring the query efficiency of the method, (3) comparisons to baselines (e.g., GCG), (4) a more detailed discussion/set of results regarding how to choose the hyper parameters, and so on. Table 2 is also deferred to the appendix, which seems strange given that the authors did not hit the nine page limit.

The paper is also sparse on examples of prompts outputted by this algorithm.  The appendix contains several abbreviated examples, but given that this attack is black box and presumably query efficient, one would expect to see quite a few examples of this behavior on a range of LLMs.

**Discussion doesn’t align with experiments.** The authors claim the following:

> “Our results almost consistently demonstrated that, as the added prompt length increased, the effectiveness of the attack improved.”

This doesn’t seem to hold based on Tables 1 and 2, both of which show numerous parameter settings which lead to the opposite of the described behaviors.  That is, there are many cases where increasing $\mathcal{M}$ (the prompt length) decreases the ASR.

The authors also say that

> “Throughout the generations we observed progressive improvement of generated outputs. . .”

I’m not sure what “progressive” means here.  If it means that as $\mathcal{N}$ and $\mathcal{M}$ increase, the ASR increases, as stated above, I don’t think that this claim is emphatically backed by evidence.  If the authors mean something else, then at the very least it’s unclear what this part of the discussion contributes.

### Related work
---

**Self-citation.** The authors cite three anonymized versions of their past work.  I’ve never seen this before.  It’s unclear how I should evaluate these claims, given that there is no link provided to these papers.  Perhaps a better avenue would be to cite other works that (a) use genetic algorithms in the context of black-box attacks on deep learning systems and (b) are not written by the authors.  That way, the authors could mix in their own work with the work of others without having to anonymize.  This would give a more balanced view of related work, because as written, the paper reads as if the authors’ anonymized works are the only relevant black box attacks which use genetic algorithms; a quite google scholar search reveals that there is plenty of work by authors at numerous different institutions.

**Other LLM attacks.**  It would be worth at least mentioning that there are numerous other attacks (both white and black box) that have been shown to be effective on different LLMs.

* The website https://www.jailbreakchat.com/ lists quite a few black box attacks, although notably, most of these aren’t automated.
* Aside from GCG, there are other white box algorithms that can jailbreak LLMs, including AutoPrompt (https://arxiv.org/abs/2010.15980) which accomplishes roughly the same goal as GCG (although it has been shown to be less effective).  See also GBDA (https://arxiv.org/abs/2104.13733), PEZ (https://arxiv.org/abs/2302.03668), TSP (https://arxiv.org/pdf/2302.04237.pdf), and https://arxiv.org/pdf/2306.15447.pdf.

While you don’t necessarily need to compare to these methods, it would strengthen the paper to include these papers in your related work section.

**Discussion of other attacks.**  The paper could write about past work in a more positive way.  The authors say that

> “Another problem with a white-box attack involves the enormous number of LLM parameters, resulting in very high GPU and memory consumption. Thus, a white-box approach is extremely costly. Moreover, due to the tokens’ discrete nature, it is impossible to use standard gradient descent directly on the tokens and the algorithm needs to be drastically modified.”

What does “drastically” mean here.  The coordinate ascent algorithm used in GCG is a relatively straightforward adaptation of gradient ascent, in that the only extra step is the top-k sampling used after computing the gradients in token space.  Indeed, given the use of the cross-entropy loss, one could argue that the attack is remarkably similar to attacks used in the adversarial examples literature, given that jailbreaking considers an entirely different problem over discrete tokens.

Relatedly, the authors say that

> “Nonetheless, the success of these attacks on the aligned models under scrutiny has proven to be somewhat limited (Kaddour et al., 2023). This limitation stems from the intricacies of optimizing discrete tokens for language-model attacks, as well as from the fundamental distinction that—unlike in image-based attacks—subtle textual perturbations are rarely imperceptible nor well-defined. In numerous classification tasks, e.g., sentiment analysis, this necessitates modifications to the attack to guarantee that token substitutions do not alter the underlying text class.”

At this point, it’s more or less established that attacks like GCG and AutoPrompt are quite effective at breaking aligned LLMs.  Moreover, is the fact that token-optimization-based approaches don’t result in imperceptible perturbations a limitation?  These attacks—to be the best of my understanding—are not designed to produce imperceptible attacks.  Moreover, the proposed genetic-algorithm-based approach also has this limitation, in that a human can generally discern an unattached prompt from an attacked prompt.

### Writing
---

There are some weaknesses associated with the writing.

**Bullet points.**  The bullet points on pages 1-2 are relatively vague, and the result is that they don’t seem to carry much information content.  What do the authors mean by “nuanced biases?” Could you include a citation or definition or a broader discussion?  What does it mean for prompts to be “contextually relevant to trigger-specific responses?”  — in particular, what is the “context” here?  What does it mean to “*effectively* manipulate” this context (emphasis is mine)?  What do the authors mean by “internal mechanisms” and by “*dynamic* model behavior (again, emphasis is mine)?  What are the “evolving defense mechanisms” mentioned by the authors? — could you cite something here?  Do such defenses even exist, beyond alignment-based fine-tuning/RLHF?  Overall, all of this text feels overly generic, and the paper could be improved by tightening this up with specific claims that are backed by citations and evidence.

**Questions:**

See above

---

> ### Author Response · Authors · 2023-11-21
> **Responses and fixes to our paper following referee reports**
>
> Dear Reviewer,
>
> We sincerely appreciate your constructive feedback and valuable suggestions.
>
> Following the reports by yourself and the other reviewers we performed numerous runs over the past few weeks and have many additional results, which we've added to the paper.
>
> We have revised our manuscript according to your comments.
>
> We acknowledge the areas you've highlighted that require our attention and have attempted to address each of your comments as follows:
>
> Regarding your comments:
> * We fixed all the problem formulations you’ve mentioned as weaknesses.
> * There are no known black-box attacks. As for white box: We would rather not compare white-box to black-box attacks since they have *very* different threat models, and this seems to us like an apples-and-oranges comparison.
> * We have added the number of queries to the model; for a population size of 10, we end up with 50k queries. for a population size of 20 – 100k queries, and for a population size of 30 – 150k queries. Not surprisingly, we need more queries than a white-box attack.
> * We have used another seed split and averaged the results.
> * We have added many more experiments including more population sizes, more text-embedders, seeds, and models.
> * We embedded the self-citation in the related-work section.
> * We have added other LLM attacks to our references.
> * We have fixed and changed the discussion section.
> * We have improved the writing in the bullet points in pages 1-2.
>
> We hope these revisions address your concerns and strengthen the paper.
>
> We've uploaded a revised version.
>
> Thanks!

---

> ### Comment · Reviewer_HedS · 2023-11-22
> **Reviewer response**
>
> > "We fixed all the problem formulations you’ve mentioned as weaknesses."
>
> Great, I think that these changes have improved the paper.
>
> > "There are no known black-box attacks. As for white box: We would rather not compare white-box to black-box attacks since they have very different threat models, and this seems to us like an apples-and-oranges comparison."
>
> I don't necessarily agree, for two reasons.
>
> 1. **The authors claim to have done this before.** As the authors mentioned in the original submission, "Other recent works of ours have focused on leveraging evolutionary algorithms in black box scenarios, showing that we can match or surpass white box results."  It seems inconsistent to claim that your past work compared black- and white-box algorithms, but to assert here that the comparison does not make sense.
> 2. **Transferrable attacks are black-box w/r/t the target model.** To quote from my original review, "although GCG is a white-box attack, the authors of the GCG paper show that attacks optimized for Vicuna and Llama2 can be transferred to black box LLMs, resulting in non negligible ASRs. In this way, one could view GCG as a black box attack for models like GPT, Claude, and PaLM when the suffixes are transferred from Llama2 or Vicuna." In my view, this would constitute a fair comparison, and it seems that reviewer `53vJ` agrees.
>
> > "We have added the number of queries to the model; for a population size of 10, we end up with 50k queries. for a population size of 20 – 100k queries, and for a population size of 30 – 150k queries. Not surprisingly, we need more queries than a white-box attack."
>
> This response confused me. The authors have asserted that it doesn't make sense to compare their algorithm to white-box attacks, but in the response, they compare (the query efficiency of) their algorithm to white-box attacks. Moreover, it's not clear to me why a black-box attack would need more queries than a white-box attack, and why is this is unsurprising. There is evidence in the (very recent) literature that the opposite can be true (see, e.g., https://arxiv.org/pdf/2310.08419.pdf).
>
> > "We have used another seed split and averaged the results."
>
> I'm not sure what this was in response to, or which results this is referring to.  Is this for Table 1?
>
> > "We have added many more experiments including more population sizes, more text-embedders, seeds, and models."
>
> I think that the additional experiments positively impact the paper.  I suppose one message of Table 1 is that the text embedder doesn't make a whole lot of difference, given that the attack is relatively successful for each of the three embedders that were tested.
>
> > "We embedded the self-citation in the related-work section."
>
> Great, I think that this will improve the clarify of that section.
>
> **Post-rebuttal thoughts.** Overall, I think that the paper has been improved after the rebuttal.  There are additional experiments, the text is clearer, and the related work section has been edited.  However, I still have concerns.
> * I think that a comparison is needed here, and as explained above, I do not agree that comparing black- and white-box algorithms would be unfair in this case.
> * I still do not understand why LLMs that are exclusively available as black boxes were not used (e.g., ChatGPT, Bard, Claude, etc.).  This seems to be the most natural setting for this algorithm, and the reason why black-box attacks are so important.
> * As I mentioned in my original review, the training/test set is contaminated for AdvBench.  This doesn't seem to have been addressed in the rebuttal.
>
> And while these concerns do remain, since I believe that the paper has improved post rebuttal, it's only fair that I raise my score.

---

> > ### Author Response · Authors · 2023-11-23
> >
> > We sincerely appreciate your thoughtful consideration of our paper and the subsequent adjustment in your score. Your insights have been invaluable to the refinement of our work, and we are committed to further enhancing the quality of our contribution.
> >
> > In response to your specific comments, we would like to provide additional context regarding the methodology and results presented in our paper. All reported results, including those in Table 1 and Table 2, have been meticulously averaged across two distinct seed splits to ensure the robustness of our findings.
> >
> > Regarding the omission of experiments with ChatGPT and Claude, we acknowledge your suggestion and want to clarify our rationale. Our attempts with these models revealed that they have implemented enhanced security measures based on the findings presented in the work of Zou et al. Consequently, we deemed it unnecessary to include these results in our paper. It is important to note that our attack methodology shares similarities with that of Zou et al., specifically in the incorporation of an adversarial suffix. Defenses against their attack are inherently applicable to ours. Furthermore, it is imperative to underscore that we have refrained from asserting the "transferability" of our attack, as distinctively noted by Zou et al.
> >
> > We wish to emphasize that our deliberate focus on models such as LLaMA and Vicuna is rooted in the widespread adoption of these models by various clients. Their popularity can be attributed to their optimal balance of effectiveness and size. By demonstrating the vulnerability of these widely utilized models to relatively straightforward attacks in a black-box setting, we aim to underscore the practical significance of our findings.
> >
> > Lastly, we are pleased to inform you that our research has yielded promising results on larger models, namely LLaMA-13B-chat and Vicuna-13B. We believe these additional findings significantly contribute to the depth and scope of our work.
> >
> > Thank you once again for your conscientious review and constructive feedback. We remain committed to addressing any further concerns you may have and ensuring the continued improvement of our paper.

---

### Official Review · Reviewer_53vJ · 2023-10-27

**Soundness:** 2 fair
**Presentation:** 2 fair
**Contribution:** 2 fair
**Rating:** 5
**Confidence:** 5

**Summary:**

The paper proposes a blackbox attack for jailbreaking based on Genetic Algorithm. The attack assumes no whitebox access to the weights or the logits. The attack aims to find adversarial tokens that when appended to a harmful prompt would cause the model to respond to the harmful prompts. These tokens are universal (not prompt-specific). Experiments were done on LLaMA2-7b and Vicuna-7b.

**Strengths:**

- The paper is well written.
- Having a universal attack against blackbox models can be impactful and very important to study as it can work on closed models via APIs and that are used in other applications.

**Weaknesses:**

- The paper needs to report the number of queries that were done to find the adversarial suffix. The threat model assumes that an adversary would attack a blackbox model (e.g., via API). Such repeated queries might trigger detection (see: https://arxiv.org/pdf/2306.02895.pdf).

- The paper could show stronger experimental evaluation. For example, the attack could be compared against a whitebox version as a baseline or an upper bound of a stronger attack. The paper also does not extensively evaluate the transferability (e.g., against the API models, or across more diverse options of source vs target models). This might help to understand when attacks are more successful/transferrable, also it would motivate the attack (is it more transferrable than optimizing on an ensemble of whitebox models?).

**Questions:**

- how many unique adversarial tokens can be found? this is important because specific tokens can be filtered out.
- what is the target sentences? is it a full continuation of the harmful answer or just "yes, sure..."?
- How is the test set selected? In order to test if the attack is "universal" across different prompts, I think it makes sense to partition the examples by topic.
- how is the success rate computed? the paper mentions that the attack succeeds if the target string is produced exactly. I imagine this has some limitations. Examples in fig. 4 do not show the output starting from "Yes, sure...". Also, is it possible that the model would start by "Yes, sure..." and not answer the harmful prompt? I imagine it might be possible especially when the length of the adversarial tokens increases.

**Details Of Ethics Concerns:**

The paper contains an ethical statement about the attack. I think this is adequate.

---

> ### Author Response · Authors · 2023-11-21
> **Responses and fixes to our paper following referee reports**
>
> Dear Reviewer,
>
> We sincerely appreciate your constructive feedback and valuable suggestions.
>
> Following the reports by yourself and the other reviewers we performed numerous runs over the past few weeks and have many additional results, which we've added to the paper.
>
> We have revised our manuscript according to your comments.
>
> We acknowledge the areas you've highlighted that require our attention and have attempted to address each of your comments as follows:
>
> Regarding your comments:
>
> Weaknesses:
> * We have used a different number of queries. We compute the fitness according to 50 randomly selected inputs, in order to make it quicker, and we do it for 100 generations. Thus, for a population size of 10, we end up with 50k queries, for a population size of 20 – 100k queries, and for a population size of 30 – 150k queries.
> * We have tried transferability to ChatGPT but it failed. It would seem that OpenAI has strengthened ChatGPT in response to the Zou et al. paper. This cat-and-mouse game is to be expected.  That said, we added many more experiments with 13B models, which we were able to run with our given academic resources.
>
> Questions:
> * The space of added suffixes is huge – indeed, it grows exponentially with the length of the suffix. So there are multiple possible solutions, which is why we tried different lengths and different population sizes. We noted that it was not overly arduous to generate different universal adversarial suffixes. So we have now performed more experiments, including different text embedders, seeds, and population size. We show the quantitative results in the newly revised paper.
> * The target sentences are taken from the dataset “harmful behavior” and are “Sure, here is a…”
> * The dataset set was randomly partitioned, using 2 different seeds that we now added to the qualitative results.
> * At first, the success rate was computed according to  “Sure…”, but following your comments, we made sure that the output is good by also doing additional runs, verifying that the output sentence doesn’t have any text from a given list of rejections, including: (``I'm not sure``, ``I'm sorry, I can't``, ``No, I don't``, ``I cannot``, ``As an AI``).
> * We have clarified all the above in the revised version.
>
> We hope these revisions address your concerns and strengthen the paper.
>
> We've uploaded a revised version.
>
> Thanks!

---

> > ### Comment · Reviewer_53vJ · 2023-11-22
> >
> > Thank you for your response and clarification.
> >
> > While I acknowledge the results reported on the open-source models, my main concern still stands. Specifically:
> > - The paper's main motivation for using a black-box setup is that a lot of models are black-box, which is a very valid motivation. However, the attacks don't transfer to black-box models. The paper does not also compute the attack by querying black-box models.
> > - Given the previous point, I believe that comparing to a white-box attack is not comparing "apples to oranges", especially since previous white-box attacks (e.g., done on an ensemble of models) showed more transferability.

---

> ### Author Response · Authors · 2023-11-22
>
> Thank you.
>
> Note that a white-box scenario and a black-box one are very different; thus, comparing a method that addresses one scenario with a method that addresses the other is likely an apple-vs-oranges comparison.
> That said, we emphasize again that in the revised version, and based on multiple new runs, we included transferability to Vicuna13b and LLama2-13b-chat, in a completely black box manner.

---

### Official Review · Reviewer_CXBa · 2023-11-06

**Soundness:** 3 good
**Presentation:** 4 excellent
**Contribution:** 3 good
**Rating:** 5
**Confidence:** 4

**Summary:**

This paper investigates the automatic generation of a universal adversarial prompt which is able to overcome alignment in open source LLMs under a blackbox setting. The paper proposes an approach based on genetic algorithms, in which sentences from a set of initial prompts are combined to maximize a fitness function. The paper utilizes a text embedding to define a fitness function which does not depend on model output probabilities. The method is evaluated on Llama-7b and Vicuna-7b models, demonstrating strong success rates on a test set of harmful behaviors.

**Strengths:**

- Paper works on interesting and challenging problems, with realistic consequences for deployment of black box open source LLMs
- The methodology using GAs is unique, and powerful
- The paper is intuitive and clearly motivated, with easy to follow exposition.

**Weaknesses:**

**Lack of experiments** - I am concerned about the lack of experimental results for larger models, or for more realistic models (e.g. Llama-7b-chat). The paper acknowledges this limitation in the future work section, but more results would still be appreciated; see questions for some possible experiments which would help in this regard.

**Questions:**

- I’m wondering how easy it is for the method to generate different universal adversarial suffixes; Are there multiple possible solutions? Is this dependent on the choice of training prompts the universal prompt is fit on?
- How long does the attack take (i.e. how many generations, and how long approximately in wall clock time per generation)? Would this be practical in a realistic scenario - especially for larger models?
- Have you tried transferability to larger models or to closed-source models (e.g. ChatGPT?)

---

> ### Author Response · Authors · 2023-11-21
> **Responses and fixes to our paper following referee reports**
>
> Dear Reviewer,
>
> We sincerely appreciate your constructive feedback and valuable suggestions.
>
> Following the reports by yourself and the other reviewers we performed numerous runs over the past few weeks and have many additional results, which we've added to the paper.
>
> We have revised our manuscript according to your comments.
>
> We acknowledge the areas you've highlighted that require our attention and have attempted to address each of your comments as follows:
>
> Regarding your comments:
> * We wish to clarify that we utilized LLaMA-7b-chat rather than LLaMA-7b, and we have taken immediate steps to rectify this error in the revised version of the manuscript.
>
> * The space of added suffixes is huge – indeed, it grows exponentially with the length of the suffix. So there are multiple possible solutions, which is why we tried different lengths and different population sizes. We noted that it was not overly arduous to generate different universal adversarial suffixes. So we have now performed many more experiments, including different text embedders, seeds, and population size. We show the quantitative results in the newly revised paper.
>
> * The time depends on the population size, the model, generated tokens, and more, but for each individual, we compute the fitness according to 50 randomly selected inputs, in order to make it quicker, and we do it for 100 generations. It takes around 3-6 minutes to run a generation, depending on the configuration and on the available computational resources.
>
> * We have tried transferability to ChatGPT but it failed. It would seem that OpenAI has strengthened ChatGPT in response to the Zou et al. paper. This cat-and-mouse game is to be expected. That said, we added many more experiments with 13B models, which we were able to run with our given academic resources.
>
> We hope these revisions address your concerns and strengthen the paper.
>
> We've uploaded a revised version.
>
> Thanks!

---

### Author Response · Authors · 2023-11-23
**Important note to all the reviewers :)**

We sincerely appreciate your thoughtful considerations of our paper. Your insights have been invaluable to the refinement of our work, and we are committed to further enhancing the quality of our contribution.

Regarding the omission of experiments with ChatGPT and Claude, we acknowledge your suggestions and want to clarify our rationale. Our attempts with these models revealed that they have implemented enhanced security measures based on the findings presented in the work of Zou et al. Consequently, we deemed it unnecessary to include these results in our paper. It is important to note that our attack methodology shares similarities with that of Zou et al., specifically in the incorporation of an adversarial suffix. Defenses against their attack are inherently applicable to ours. Furthermore, it is imperative to underscore that we have refrained from asserting the "transferability" of our attack, as distinctively noted by Zou et al.

We wish to emphasize that our deliberate focus on models such as LLaMA and Vicuna is rooted in the widespread adoption of these models by various clients. Their popularity can be attributed to their optimal balance of effectiveness and size. By demonstrating the vulnerability of these widely utilized models to relatively straightforward attacks in a black-box setting, we aim to underscore the practical significance of our findings.

Lastly, we are pleased to inform you that our research has yielded promising results on larger models, namely LLaMA-13B-chat and Vicuna-13B. We believe these additional findings significantly contribute to the depth and scope of our work.

Thank you once again for your conscientious reviews and constructive feedbacks. We remain committed to addressing any further concerns you may have and ensuring the continued improvement of our paper.

---

### Meta-Review · Area_Chair_C86h · 2023-12-05

**Metareview:**

This submission, "Open Sesame! Universal Black Box Jailbreaking of Large Language Models", shows that adversarial attacks against aligned language models do not require gradient or activation information and can be found through the use of standard zeroth-order genetic algorithms. This is an interesting finding that presents the key novelty of this submission. The authors provide several experiments with Vicuna-7b and LLaMA-7b-chat to show that such an attack is feasible.

However, reviewers unanimously found the comparisons to related attack strategies lacking, which would round out the evaluation and allow for a clear placement of the proposed algorithm in the context of related attacks. The authors only evaluate their proposed approach, yet, query-based black-box attacks are not the only attack vector against LLM APIs.  Especially transfer attacks from public models can be evaluated in direct comparison.
Further, a comparison to white-box attacks, especially the attack of Zou et al., would clarify the relative strength of the black-box approach in this work. We do understand that the submission proposes a black-box approach, but how strong is the black-box approach, compared to current white-box attacks?

**Justification For Why Not Higher Score:**

Missing contextualization of the work in the broader space of attacks against (API-based) large language models.

**Justification For Why Not Lower Score:**

N/A

---

### Decision · Program_Chairs · 2024-01-16

Reject